# Advancing Newborn Screening in Washington State: A Novel Multiplexed LC-MS/MS Proteomic Assay for Wilson Disease and Inborn Errors of Immunity

**DOI:** 10.3390/ijns11010006

**Published:** 2025-01-10

**Authors:** Claire Klippel, Jiwoon Park, Sean Sandin, Tara M. L. Winstone, Xue Chen, Dennis Orton, Aranjeet Singh, Jonathan D. Hill, Tareq K. Shahbal, Emily Hamacher, Brandon Officer, John Thompson, Phi Duong, Tim Grotzer, Si Houn Hahn

**Affiliations:** 1Key Proteo, Inc., Seattle, WA 98122, USA; claire.klippel@keyproteo.com (C.K.); jiwoon.park@keyproteo.com (J.P.); sean.sandin@keyproteo.com (S.S.); 2Alberta Precision Laboratories, Calgary, AB T2L 2K8, Canada; tara.winstone@albertaprecisionlabs.ca (T.M.L.W.); xue.chen@albertaprecisionlabs.ca (X.C.); dennis.orton@cw.bc.ca (D.O.); 3Washington State Department of Health Newborn Screening Laboratories, Seattle, WA 98155, USA; aranjeet.singh@doh.wa.gov (A.S.); jonathan.hill@doh.wa.gov (J.D.H.); tareq.shahbal@doh.wa.gov (T.K.S.); emily.hamacher@doh.wa.gov (E.H.); brandon.officer@doh.wa.gov (B.O.); john.thompson@doh.wa.gov (J.T.); 4Seattle Children’s Research Institute, Seattle, WA 98105, USA; phi.duong@seattlechildrens.org (P.D.); tim.grotzer@seattlechildrens.org (T.G.)

**Keywords:** Wilson Disease, Wiskott–Aldrich syndrome, X-linked agammaglobulinemia, adenosine deaminase deficiency, newborn screening, proteomics, peptide, mass spectrometry

## Abstract

For many genetic disorders, there are no specific metabolic biomarkers nor analytical methods suitable for newborn population screening, even where highly effective preemptive treatments are available. The direct measurement of signature peptides as a surrogate marker for the protein in dried blood spots (DBSs) has been shown to successfully identify patients with Wilson Disease (WD) and three life-threatening inborn errors of immunity, X-linked agammaglobulinemia (XLA), Wiskott–Aldrich syndrome (WAS), and adenosine deaminase deficiency (ADAD). A novel proteomic-based multiplex assay to detect these four conditions from DBS using high-throughput LC-MS/MS was developed and validated. The clinical validation results showed that the assay can accurately identify patients of targeted disorders from controls. Additionally, 30,024 newborn DBS samples from the Washington State Department of Health Newborn Screening Laboratory have been screened from 2022 to 2024. One true presumptive positive case of WD was found along with three false positive cases. Five false positives for WAS were detected, but all of them were premature and/or low-birth-weight babies and four of them had insufficient DNA for confirmation. The pilot study demonstrates the feasibility and effectiveness of utilizing this multiplexed proteomic assay for newborn screening.

## 1. Introduction

The successful introduction of newborn screening (NBS) for phenylketonuria in dried blood spots (DBSs) by the pioneering work of Dr. Robert Guthrie in the early 1960s has been the catalyst for establishing population-based newborn screening in the United States for over 30 disorders to date [1]. There has been increasing global interest in expanding newborn screening panels to include more genetic diseases due to significant advances in recent treatment options, particularly cell- and gene-based modality, and the early detection of these diseases allows patients to receive these life-saving preemptive treatments [1]. However, the efficacy of early treatment is largely limited by the lack of available early diagnostic methods. An international working group published an essential list of medicinal products for rare diseases in which 134 diseases had available treatments, and 61 of those were protein deficiency diseases without available NBS methods [2]. As most causative mutations in genetic disorders result in a reduction in or absence of target proteins, directly measuring the respective peptides as surrogate biomarkers of those intracellular proteins can lead to an effective diagnosis [3,4]. Rapid advances in proteomic techniques have enabled the detection and quantification of targeted peptides in a very limited sample volume such as dried blood spots [5].

Proteomic immuno-SRM (selected reaction monitoring) is used to screen for specific genetic disorders by quantifying extremely-low-abundance proteins from blood samples that include both serum and intracellular proteins [3,4,6,7,8]. Both serum and intracellular proteins can be quantified directly from patient DBS samples by extracting and digesting proteins, adding known amounts of stable isotope-labeled standards, using magnetic antibody beads for the immunoaffinity enrichment of target proteolytic peptides, and then quantifying the target peptides by liquid chromatography-tandem mass spectrometry (LC-MS/MS) [3,4].

Multiplex assays were first developed by Chace and Millington in the late 1990s to simultaneously analyze multiple small abundant molecules using tandem mass spectrometry [9]. This technology is particularly useful for large-scale population-based screening, as it reduces sample analysis time, increases sample throughput, and lowers operational costs per sample [10]. Multiplexed proteomic immuno-SRM methods have been recently developed to screen newborns for Wilson Disease, X-linked agammaglobulinemia, Wiskott–Aldrich syndrome, and adenosine deaminase deficiency from dried blood spots [3,4,11,12,13].

Wilson Disease (WD) is a copper transport disorder caused by a mutation in the *ATP7B* gene, with an estimated incidence rate of 1 in 30,000 and a higher carrier frequency of 1 in 90 [14]. WD is asymptomatic at birth, and if left untreated, patients develop irreversible hepatic, neurologic, psychiatric disturbances, or a combination of these between the ages of three years to older than 70 years [14,15]. Early treatment with chelating agents or zinc salts can greatly improve their prognosis and quality of life [16]. Previously, ceruloplasmin was explored as a potential biomarker for the newborn screening of Wilson Disease, but proved ineffective due to a high false positive rate in newborns [17,18].

Wiskott–Aldrich syndrome (WAS), X-Linked agammaglobulinemia (XLA), and adenosine deaminase (ADA) deficiency are inborn errors of immunity (IEI), a group of over 485 genetic disorders that compromise immunity due to an improperly functioning immune system [19,20]. XLA is caused by a genetic defect of Bruton’s tyrosine kinase (BTK) protein, with an incidence rate of 1 in 200,000 [21,22]. This disorder is characterized by a severe B-cell deficiency, low immunoglobin levels, and recurrent bacterial infections in affected males in the first two years of life, and most patients are treated with immunoglobulin replacement therapy or prophylactic antibiotics [22,23,24,25]. WAS is a combined immunodeficiency disorder characterized by thrombocytopenia, and it primarily affects males with dysfunctional platelets due to defects in the WAS protein (WASp) that lead to bleeding problems and severe recurrent infections, with an incidence rate of between 1 and 10 in 1 million [26,27,28]. WAS patients are treated with hematopoietic stem cell transplantation (HSCT) [29,30]. ADA deficiency is caused by mutations in the ADA gene and affects lymphocyte development and function [31,32]. About 80% of patients manifest severe combined immunodeficiency (SCID) in their infancy, while the remaining 20% develop late-onset combined immunodeficiency (CID) later in life [32]. While there is currently screening for early-onset ADA deficiency [33], a small percentage of patients who develop symptoms later in life can be missed by this NBS method [34,35]. Enzyme replacement therapy, gene therapy or HSCT are the available treatment options [36,37].

In previous studies, all IEI patients showed absent or significantly reduced target peptide levels in dried blood spots (WAS, n = 18; XLA, n = 37) [3,13]. The early detection of IEI is essential to manage and prevent potentially life-threatening infections and other severe disease complications [38]. Currently, IEI in newborns are most commonly detected using the T-cell receptor excision circle (TREC) assay for T-cell deficiencies or the kappa-deleting excision circle (KREC) assay for B-cell deficiencies [39,40]. As TREC testing does not target specific conditions [39], many costly additional tests are necessary for the final confirmation of a diagnosis, and unwanted complications may emerge before the final diagnosis [41,42]. Flow cytometry and DNA sequencing are the top analytical methods for confirming the initial abnormal newborn screening for a final diagnosis [43].

In this study, we present a novel multiplexed proteomic assay that measures six peptides to screen for WD, XLA, WAS, and ADA deficiency. The multiplex assay includes a total of six peptides selected based on previous studies that demonstrated these biomarkers are detectable using LC/MS-MS instrumentation and elicited specific antibodies for isolation [3,12,13]. Specifically, two peptides were selected for both Wilson Disease and X-Linked agammaglobulinemia to ensure an accurate detection and build assay sensitivity, while one peptide was selected for both Wiskott–Aldrich syndrome and ADA deficiency. The immuno-SRM assay has been validated for analytical and clinical precision following the Clinical and Laboratory Standards Institute (CLSI) guidelines. Additionally, we completed a large-scale pilot study of 30,000 newborn DBS samples from the Washington State Department of Health Newborn Screening Laboratory to demonstrate the feasibility of integrating this assay into the current public NBS panel in Washington State and beyond.

## 2. Materials and Methods

### 2.1. Validation

The samples for all validation studies were constructed from a mixture of human blood and fish blood purchased from BioIVT (Westbury, NY, USA). Fish blood was used as a negative control as it lacks human homologs of all six targeted proteins. Red blood cells and plasma were mixed in equivalent volumes to produce human and fish blood with ~55% hematocrit, similar to levels in newborns [44]. Both fish blood and human blood were spotted onto blank filter papers and dried overnight to mimic actual dried blood spot samples. Assay validation studies were designed following CLSI guidelines [45,46,47,48,49,50,51]. All studies included quality control samples that consisted of 0% human/100% fish blood (QC negative), 50% human/50% fish blood (QC low), and 100% human/0% fish blood (QC high). Quality control samples were accepted when within previously established concentration ranges.

#### 2.1.1. Analytical Sensitivity

The linearity intervals of the six peptides were determined by spiking light peptide standards (Thermo Fisher Scientific, Waltham, MA, USA) into 100% fish blood to create twelve concentration levels listed in Appendix A. Five replicates of each concentration level were used. A simple linear regression was used for each peptide to determine the lower and upper limit of the linearity interval (Appendix A). In addition, the limit of blank (LOB), limit of detection (LOD), and limit of quantification (LOQ) of the assay were determined according to the CLSI guidelines [51]. The limit of blank samples consisted of 60 replicates of SigMatrix IVD mock serum (Millipore Sigma, Burlington, MA, USA), which does not contain any screened biomarker peptides. Five replicates of 2.5%, 5%, 7.5%, and 10% human blood samples were used to determine the limit of detection, and five replicates of 2.5%, 5%, 7.5%, 10%, 20%, and 30% human blood samples were used to determine the limit of quantification.

#### 2.1.2. Interference

Both 100% fish blood and a 30% mixture of normal human control, and 70% fish blood were spiked with 11 common interfering substances (unconjugated bilirubin, conjugated bilirubin, galactose, glucose, EDTA, heparin, total protein, hemoglobin, cholesterol, triglycerides and gamma-globulin) (Appendix A). Pooled samples spiked with an equivalent amount of the control solvent from the interferent kit (Molecular Depot, San Diego, CA, USA) were used as the control group. The Student *t*-test was used to determine whether the presence of each interferent significantly affected peptide concentrations.

#### 2.1.3. Carryover

To determine the carryover potential for each peptide, blank samples were run prior to and immediately after a DBS sample with high analyte concentrations. High-analyte DBS samples were prepared by spiking 100% normal human blood with commercially obtained buffy coat (BioIVT, Westbury, NY, USA). Cells comprising an isolated buffy coat contain high concentrations of target peptides. The percent carryover was calculated by dividing the peptide concentrations in blank samples run immediately after high-analyte samples by peptide concentrations in high-analyte samples.

#### 2.1.4. Reproducibility and Precision

Three study sites were utilized to determine the repeatability, reproducibility, and between-day precision of the assay. Studies at site 1 (Alberta Precision Lab, Calgary, AL, Canada) and site 2 (Seattle Children’s Research Institute, Seattle, WA, USA) were conducted over 5 days, with two plates per day and 3 replicate measurements of identical DBS samples by two operators for a total of 360 measurements. The sample compositions included 0, 20, 40, 60, 80, and 100% normal human control blood (NHC). The study at site 3 (Key Proteo, Seattle, WA, USA) was conducted over 20 days, with 2 plates per day and 3 DBS replicates by two operators for a total of 720 measurements. All datasets were analyzed using a two-way nested ANOVA, as described in the CLSI EP05-A3 document [47].

#### 2.1.5. Clinical Validation

The screening performance, sensitivity, and specificity of the assay were determined in a prospective clinical study. The samples included 3294 routine newborn screening samples from the Washington State Department of Health Newborn Screening Laboratory and 49 previously confirmed positive de-identified patient samples stored in the Seattle Children’s Research Institute biorepository at −20 °C after IRB approval. Six of these affected patient samples were left-over original newborn blood spots retrieved from the Washington State Department of Health Newborn Screening Laboratory. The patient samples included 32 Wilson Disease patients, 8 XLA patients, 6 WAS patients and 3 ADA patients, and were divided between three sites to be run blinded along with routine samples. Sites 1 and 2 each ran 601 routine samples in addition to 15 and 14 positive controls, respectively, representing each of the four target diseases. Site 3 ran 2092 routine samples in addition to 20 positive controls for the same four diseases.

To establish an initial diagnostic cutoff value for each peptide, a preliminary study was performed by analyzing 1056 presumed negative de-identified newborn samples provided by the Washington State Department of Health Newborn Screening Laboratory and 12 positive control samples. For each screen-positive and screen-negative group, the mean, median, and standard deviation of each peptide concentration were calculated. The initial diagnostic cutoffs were set at 10% of the cohort median for BTK and WASP peptides, 25% for ATP7B peptides, and 20% for the ADA peptide. The cutoffs were selected to ensure that all known affected specimens were detected and minimize false positive rates. Any sample that was identified as a potential positive case based on these cutoffs was subjected to genetic sequencing. Additionally, to evaluate the possibility of false negative cases (those who showed normal assay results but are actually affected individuals), 100 borderline abnormal samples (34 for the *ATP* gene, 32 for the *BTK* gene, 16 for the *WAS* gene, and 18 for the *ADA* gene) with peptide concentrations 20% or higher than the median were selected and sent out for the sequencing of the target genes.

#### 2.1.6. Stability

All reagents necessary for the multiplex assay were generated at the Argonaut Manufacturing Services facility (Carlsbad, CA, USA) under GMP conditions and provided as a screening kit for investigational use. To determine the stability of these kit components, all components were stored at −20 °C and used to process 0% human blood/100% fish blood, 50% human blood/50% fish blood, and 100% human blood/0% fish blood specimens at 0, 3, 6, 9, 12, and 13 months from the time of receipt. In addition, to determine the stability of DBS samples over time, 100% human blood DBS cards were stored at 3 different temperatures (10 °C, 25 °C, 37 °C) and were processed daily over 5 days. For both stability experiments, bias over time was calculated as the percent difference in peptide concentrations between 0 months and each timepoint.

### 2.2. Pilot Study

The project titled “Pilot Study for Newborn Screening of Wilson Disease and IEI (XLA, WAS, and ADAD)” (2021-085-Department of Health) was approved by the Washington State Institutional Review Board. This study was conducted at three approved sites: Key Proteo Inc., Seattle Children’s Research Institute (IRB STUDY00003814), and Alberta Precision Laboratories. The pilot study utilized de-identified remnant DBS provided by the Washington State Department of Health Newborn Screening Laboratory after routine newborn screening. It was determined that the study poses no greater than minimal risk (45 CFR 46.404/21 CFR 50.51). The study also involves a diagnostic device that is IDE-exempt under 21 CFR 812.2(c) (3), allowing the study team to distribute and use medical device (the screening kit) that have not yet been approved by the FDA. Additionally, waivers of consent/assent and parent/guardian permission for study participation were granted in accordance with 45 CFR 46.408(c). Duplicates of 96-well plates containing de-identified newborn DBS samples were received by Key Proteo, each to be used as an experimental plate and a genetic sequencing plate. Both plates contained 88 pre-punched individual samples, 3 punches per well in the experimental plate and 6 punches per well in the genetic sequencing plate. In the last column on the experimental 96-well plate, three 3.2 mm diameter DBS punches were inserted in each well using a Panthera Puncher (Revvity, Waltham, MA, USA). These included two wells for blanks, two wells for 0% human blood/100% fish blood (QC negative), two wells for 50% human blood/50% fish blood (QC low), and two wells for 100% human blood/0% fish blood (QC high). Additionally, to evaluate the possibility of false negative cases, 31 borderline abnormal samples (24 for the *ATP* gene, 5 for the *WAS* gene, and 2 for the *ADA* gene) were selected and sent out for the sequencing of the target genes.

### 2.3. Assay Procedure

Samples for all validation studies and pilot study were processed identically. Proteins were extracted from the DBS samples using 200 μL of 0.1% Triton X-100 in 50 mM ammonium bicarbonate and 6 μL of 0.2 M dithiothreitol and incubated for 30 min at 37 °C while shaking. Then, 37.5 μg of trypsin was added and incubated for 2 h at 37 °C while shaking to digest the proteins. The supernatant was transferred to a new 96-well plate containing known concentrations of lyophilized six heavy isotope-labeled internal standard peptides. Next, 10 μL of TRIS buffer and 18 μL of monoclonal antibody (mAb) bead solution were added to the supernatant, and the mixture was incubated overnight at 4 °C with shaking. The following day, mAb beads were collected using Alpaqua Magnetic Plate (Alpaqua, Beverly, MA, USA) and washed twice with 220 μL of PBS buffer before adding 30 μL of the elution solution (5% acetic acid + 3% ACN). The eluents were transferred to 200 μL 96-well plates for LC-MS/MS analysis.

### 2.4. LC-MS/MS Analysis

All samples for the validation studies and 19,526 pilot study samples were analyzed using a Waters Xevo TQ-XS with low flow ESI probe and dual M-class chromatography pumps. Peptides were initially loaded onto a nanoEase Symmetry C18 trap column (300 µm × 25 mm, 5 µm particle size; Waters Corporation, Milford, MA, USA) for 3 min at 40 μL/min with 98% solvent A and 2% solvent B, then eluted on a BEH-C18 analytical column (300 µm × 50 mm, 1.7 µm particle size; Waters Corporation, Milford, MA, USA) for 2.4 min at 8 μL/min (Solvent A: H_2_O + 0.1% formic acid, solvent B: ACN + 0.1% formic acid). The analytical column temperature was 75 °C. MS analysis was performed in a positive ion mode using selective reaction monitoring (SRM), with both Q1 and Q3 quadrupoles set at a unit resolution. The precursor mass, fragment mass, cone voltages, and collision energies for each peptide were adapted from previous studies [3,12,13]. Then, 10,498 pilot study samples were analyzed using a Waters Xevo TQ-XS with ionKey probe and dual M-class chromatography pumps. The liquid chromatography conditions and MS settings were identical to those described in the previous study [13].

### 2.5. Genetic Testing

Six DBS samples from the backup plates were sent to Revvity Omics Laboratory (Pittsburgh, PA, USA) for genetic sequencing when one or more peptide concentrations were below the diagnostic cutoffs.

### 2.6. Data Analysis

SRM data for pilot study were analyzed using Skyline (MacCoss Lab, Seattle, WA, USA https://skyline.ms/project/home/software/Skyline/begin.view, accessed on 22 January 2022) [52], and SRM data for validation studies were analyzed using TargetLynx (Waters Corporation, Milford, MA, USA). For both datasets, the analyte peaks were identified based on the retention times obtained from commercial internal standards (Thermo Scientific), and both endogenous (light) and isotope-labeled (heavy) peak areas were quantified in each software. Peptide concentrations were calculated using the ratio of the peak area of light peptides in the blood to the peak area of heavy peptides added at known concentrations. The samples with peptide concentrations below their respective cutoffs were reanalyzed and sent out for genetic sequencing to confirm genetic variants.

Statistical analysis was performed using Graphpad Prism (San Diego, CA, USA, https://www.graphpad.com/ accessed on 13 November 2024). To compare peptide concentrations between genders, Welch’s *t*-test was used. To compare the peptide concentrations between different body weights, the age groups and ethnicities, a Welch ANOVA and the Games–Howell post hoc test was used.

## 3. Results

### 3.1. Assay Performance Characteristics

#### 3.1.1. Analytical Sensitivity

The LOB, LOD, and LOQ for each peptide were determined as below (Table 1).

#### 3.1.2. Interference

There was no statistically significant effect from any of the 11 interferents tested on the concentrations of all six peptides except in one instance of ADA 93, but the percent difference in ADA 93 concentration between interferent-spiked and unspiked samples was below 20% (Appendix A).

#### 3.1.3. Carryover

The carryover for all peptides except ADA 93 was below 10% of the immediately preceding the high-analyte sample (Table 2). The carryover for ADA 93 was below 20% in all but one of the seven replicates.

#### 3.1.4. Reproducibility and Precision

Using Grubbs’ test, a maximum of one outlier was identified across all replicates and concentration levels for each peptide. A total of five outliers were removed from site 1 and site 3, and six outliers were removed from site 2. Aside from one instance at site 3 for ADA 93, the total precision for 100% human blood and 80% human/20% fish blood samples at all three sites was consistently below 30% CV (Appendix A).

#### 3.1.5. Clinical Validation

Initial diagnostic cutoffs were set by running 1056 newborn samples from the Washington State Department of Health Laboratory, alongside 12 positive controls. Cutoffs were set at 10% of the cohort median for BTK and WASP peptides, 25% of the cohort median for the ATP7B peptides, and 20% of the cohort median for ADA peptide (Table 3). As patients affected by the four diseases targeted by our assay exhibited highly reduced protein levels [3,12,13], peptide concentrations below the diagnostic cutoffs were considered positives. To validate these cutoffs, a total of 3294 newborns and 49 genetically confirmed positive samples were tested across 3 sites. All confirmed 49 positive cases were screen positive, and repeats were concordant with initial results. Four confirmed Wilson Disease cases at site 3 (cases 2, 6, 7, and 11) had ATP7B 887 levels above the initial range cutoff, while the ATP7B 1056 levels were below the cutoff (Appendix A). Two Wilson Disease patients (site 2 case 1 and site 3 case 10) had ATP7B 1056 levels above the cutoff, but ATP7B 887 levels were below the cutoff. In one confirmed case of Wilson Disease (site 1 case 2), WASP 274 level was also below cutoff and subsequent sequencing of WAS gene showed a variant of uncertain significance, c.133-38C>A (Appendix A).

From the 100 borderline abnormal samples for evaluating potential false negatives, no false negative cases were detected except one possible WD case. This sample (case #13) had both ATP7B 1056 and 887 concentrations above the cutoff, and one variant that was pathogenic (p.Lys838Glu) and one variant with an uncertain significance (VUS) (p.Gly1186Asp) for the ATP7B gene (Appendix A). The clinical significance of this case was uncertain. Four additional cases (#6, 7, 20, 31) were heterozygous for known likely pathogenic or pathogenic variants, with no second pathogenic variants in the ATP7B gene, indicating these samples are most likely carriers for Wilson Disease. In addition, 6 cases (case #1, 10, 12, 21, 23, 33) were heterozygous for VUS in the ATP7B gene, with no second variants. One hemizygous VUS variant in the BTK gene (case #34) was detected. Four cases were heterozygous for VUS without second variants in the ADA (case #83, 91, 94, 97) gene, indicating possible carriers for ADA deficiency. One VUS (c.133-38C>A) in the WAS gene was detected (case #82).

#### 3.1.6. Stability

The kit components were stable for up to 12 months when stored at −20 °C (Appendix A). The CVs for all peptides were considered acceptable and bias over time was under 30%. DBS cards were stable for up to five days with no statistically significant differences between days or between temperatures for all six peptides (Table 4).

### 3.2. Pilot Study

#### 3.2.1. Demographic Information of the Study Cohort

Approximately 30,024 newborn samples were tested using the multiplex assay to date. The samples were assigned a number by the Washington State Department of Health Newborn Screening Laboratory. Demographic information, including gender, birth weight, day of blood collection after birth, and ethnicity, was recorded and reported with each sample (Table 5A–C). Of that, 2877 samples did not have any ethnicity information and were not included, while 3845 samples had multiple ethnicities.

#### 3.2.2. Peptide Concentrations Across Gender, Birth Weight, Ethnicity, and Age of Collection

The distribution of peptide concentrations for males and females for each of the six measured peptides is shown in Figure 1A. For five of the six target peptides, there were statistically significant differences in peptide concentrations between genders (*p* < 0.0001 for all peptides except ADA 93), as shown in Figure 1B.

The distribution of peptide concentrations for newborn birth weights (<1500 g, 1500 g–2500 g, >2500 g) for each of the six measured peptides is shown in Figure 2A. There were significant differences in peptide concentrations between <1500 g and >2500 g in four of the six target peptides (ATP7B 887, ATP7B 1056, WASP 274, and ADA 93), and significant differences in five of the six target peptides for 1500 g–2500 g vs. >2500 g (ATP7B 887, ATP7B 1056, WASP 274, ADA 93, and BTK 407) (*p* < 0.0001) (Figure 2B).

There were statistically significant differences found in peptide concentrations between white and black ethnic groups (ATP7B887, ATP7B 1056, WASP 274, ADA93), white and Hispanic ethnic groups (ATP7B 887, ATP7B 1056, WASP 274, ADA 93, BTK 545, and BTK 407), white and Native American ethnic groups (ATP7B 887, ATP7B 1056, and ADA93), and white and Asian ethnic groups (BTK 545 and BTK407) (Figure 3A,B).

The distribution of peptide concentration from 0 to 7+ days of collection is shown in Figure 4A. WASP 274 had no significant differences between peptide concentrations for any day of collection. ATP7B 887 had statistically significant differences in peptide concentration between days 0 and 6. ATP7B 1056 had significant differences between days 0 and 3, as well as days 0 and 5. ADA 93 showed significant differences between peptide concentrations for day 0 and days 7+. BTK 545 showed significant differences between days 0 and 4, and days 0 and 7+. BTK 407 showed significant differences between days 0 and 1, days 0 and 2, days 0 and 3, days 0 and 4, and days 0 and 5 (Figure 4B).

#### 3.2.3. Determination of the Cutoff

For the samples analyzed using the ESI low-flow probe on the mass spectrometer, the cutoffs for each peptide were established at 28% of the median for ATP7B 887 and 1056, 20% of the median for ADA 93, and 10% of the median for WASP 274, BTK 545, and BTK 407 (Table 6). For the samples analyzed using the ionKey source on the mass spectrometer, the cutoffs for each peptide were established at 19% of the median for ATP7B 887 and 1056, 20% of the median for ADA 93, and 10% of the median for WASP 274, BTK 545, and BTK 407. These cutoffs were selected to minimize false positive rates and corroborated by examining previous patient concentrations and criteria [3,12,13], and sequencing borderline and negative samples. The differences in sensitivity between the Ionkey and ESI configurations also account for some discrepancies in the cutoff determination.

#### 3.2.4. Presumptive Positives

Samples with peptide concentration below the established cutoff for at least one target peptide were deemed as presumptive positives and sent out for genetic sequencing. As X-linked agammaglobulinemia and Wiskott-Aldrich syndrome are X-linked conditions [22,26], primarily male samples are presumptive positives. One sample was determined as a tentatively true patient for WD, as it contained two variants of the *ATP7B* gene that were classified as uncertain clinical significance (not determined for cis or trans) and ATP7B 1056 concentration was below the established cutoff (Sample 1) (Table 7). However, due to the nature of the blinded samples, this patient was not available for additional follow-up. Three additional samples in the pilot study cohort deemed presumptive positives for WD had either no variants, or variants that have been previously established as benign (Table 7). The positive predictive value (PPV) for Wilson Disease was 25%, and the false positive rate was 0.001%.

All five presumptive positives found for WASP 274 were determined to be false positives. All cases were premature babies that either contained no pathogenic variants or were inconclusive as there was insufficient DNA present in the samples (Table 7). Of note, in samples 6 and 7, both BTK 545 and BTK 407 concentrations were also below 20% of calculated median, indicating these cases were likely experiencing severe lymphopenia related to prematurity (Table 7). One sample (12) was sent for the sequencing of BTK 545 and BTK 407, as BTK 545 concentration was close to the currently established cutoff. The sequencing report came back with no variants, and this sample was determined to be a false positive (Table 7).

#### 3.2.5. False Negatives

Samples with peptide concentrations above but near the cutoff were sent out for sequencing as part of a false negative study to determine if the borderline samples were being missed as negative. A total of 23 samples were sent out for ATP7B, 5 samples for WASP 274, and 3 samples for ADA 93 (Table 8). Case 22 included one likely pathogenic variant and one VUS, indicating that this sample is likely a carrier. The negative predictive value (NPV) was 96% for WD, and 100% for ADA, WAS, and XLA. The false negative rate for WD was approximately 1%.

## 4. Discussion

The analytical validation study results show that the LOB, LOD, and LOQ of all six peptides were lower than the diagnostic cutoffs (Table 1), and common interferents do not affect the peptide concentrations (Appendix A). The carryover between the samples was negligible for all peptides except ADA 93, but this would rarely affect the accuracy of the assay (~0.01% samples affected in our pilot study; see Appendix B and Table 2). The reproducibility study results show that our immuno-SRM assay can produce consistent results over days or between different users/instruments (Appendix A; see Appendix B). The assay was successfully validated by detecting all 49 genetically confirmed positive controls from a total of 3294 newborn samples across 3 study sites (Appendix A; see Appendix B). Additionally, no false negatives were detected from the 100 borderline samples, except one sample (case 13) that had one *ATP7B* gene variant that is pathogenic (p.Lys838Glu) and one VUS (p.Gly1186Asp) (Appendix A). The stability study results show that our assay reagents are stable for up to one year with all six target peptides, and dried blood spot cards were stable for up to 5 days (Table 4). Overall, the analytical and clinical validations were satisfactory to apply the assay to clinical practice especially for the high-throughput assay such as the newborn screening platform.

This is the first large-scale pilot NBS study targeting WD, XLA, WAS, and ADA deficiency by LC-MS/MS to date. We observed statistically significant differences (*p* < 0.0001) in all six peptide concentrations between genders (n = 30,024), including the X-linked conditions of WAS and XLA (Figure 1). However, it should be noted that statistical significance was likely influenced by the power of our study, as large sample sizes reduce the impact of random error and inflate statistical significance [53]. Regardless, we did not need to set separate cutoffs for each gender based on the small difference between the means for all peptides.

There were also statistically significant differences in peptide concentrations from babies with different birth weights (Figure 2). ATP7B 887, ATP7B 1056, WASP 274, and ADA 93 concentrations in babies with birth weights >2500 g were significantly different (*p* < 0.0001) than those with birth weights <1500 g and 1500–2500 g. These weight brackets are typically used to discern premature and normal-term babies [54]. A newborn’s birth weight is an important factor to consider when performing a newborn screening test, as it can affect the accuracy of the screening results [55]. As the risk of a false positive result could potentially increase with a lower birth weight, a different protocol or cutoff may need to be used [55]. In our study, we did not apply different cutoffs for low birth weights for two reasons: (1) it did not affect the accurate detection of patients in our clinical validation study and (2) the sample size for premature babies was too small relative to that of normal-weight babies. A larger-scale screening study with higher numbers of premature babies will be necessary to determine whether babies with a low birth weight will need separate diagnostic cutoffs.

There were statistically significant differences in all six peptide concentrations across all ethnicities represented (Figure 3). The prevalence of the genetic disorders screened by our assay varies between different ethnicities/geographical areas. However, the difference in the means of Asian and white population concentrations was nominal, and therefore a separate cutoff was not needed to distinguish patients. Wilson Disease patients are found more frequently in isolated populations, likely due to high rates of consanguinity in the area [56,57]. Additional studies specifically targeting regions such as the island of Crete [58] and Sardinia [59,60] may be necessary to evaluate if these populations have significantly different peptide concentrations that may affect cutoffs for accurate diagnosis. In comparison, regions like France and Hong Kong have lower occurrence rates than seen in global estimates [61,62].

The age of the baby at the time of the test is also important, as well as other factors like prematurity and transfusion status. DBS samples for newborn screening tests are typically collected 24–48 h after birth [63]. If a baby is discharged before 24 h of age, a repeat test can be performed before one week [63]. A second screening between 7 and 14 days of age is obtained in WA state to reduce false negative, particularly for hypothyroidism [64]. We have found that the age of sample collection shows statistically significant differences in BTK 407 and BTK 545 concentrations. Generally, BTK 545 concentrations decreased over time, while BTK 407 concentrations increased over time (Figure 4). Given the significant changes in BTK concentrations over time, we recommend rerunning samples that have BTK concentrations within 10% of the diagnostic cutoff and are considered borderline. Although statistically significant differences were noted between specified demographics, the differences did not influence the diagnostic accuracy of the test and therefore did not require separate cutoffs.

To detect the presumptive positive cases for each of the four diseases, we used the cutoffs established during the ongoing collection of pilot study samples. It should be noted that individual laboratories should be responsible for establishing their own cutoff for each peptide, as there will be variations in peptide concentrations between the population being tested and the instrumentation used. Established cutoffs need to be continually adjusted as the sample size increases and more patient samples are screened and tested with genetic, clinical, and laboratory analysis. Using our pilot study cutoffs (Table 6), one presumptive true positive for WD was detected out of 30,024 de-identified samples from the Washington State Department of Health Newborn Screening Laboratory (Table 7). This Hispanic male patient had two heterozygous variants of uncertain significance that were reported, p.Pro610Leu and p.Arg1224Leu. The first variant has not been reported in the literature as causative of disease to date and is extremely rare, but the second variant has been reported in an individual with suspected Willson Disease [65]. As this presumptive true positive case had an ATP7B 887 concentration below cutoff but ATP7B 1056 concentration above cutoff, it is important to use multiple signature peptides to accurately identify patients.

We found five cases in which WASP 274 concentrations were below the cutoff, but all cases were premature low-birth-weight newborns. Four of them also had insufficient DNA for sequencing (Table 8). Some false positive cases could be due to prematurity related illnesses at the time of sample collection, such as leukopenia. There was one presumptive positive sample for XLA that was below cutoff and was sent for sequencing, but no variants were detected. Overall, our assay demonstrated a very low false positive rate for IEI conditions (up to 0.0001%), which is much lower than the false positive rate in TREC assay (0.02–0.1%) [39,66,67].

In the potential false negative study using the pilot samples, we found seven samples with heterozygous genetic variants of uncertain significance detected for the *ATP7B* gene, as well as heterozygous pathogenic variants detected in five samples that were screen negative. There was one case that could be a true positive, in which two variants of uncertain significance were detected in a sample with normal peptide concentration. The first allele (p. Pro539Leu) has been reported as likely pathogenic in previous studies [12], while the second allele (p. Ser876Cys) has not been linked to WD and is only reported in South Asian populations. However, the clinical significance is yet uncertain as we were not able to confirm or follow up on this case due to the nature of de-identified sample collection. Given that the carrier frequency of WD has been estimated to be 1 in 90, the current cutoff seems appropriate to differentiate between carriers and affected individuals. Five samples were sequenced for WASP 274, and in all five samples there were no variants detected. ADA 93 concentrations in all samples were above the cutoff, so we sequenced three samples that had lowest concentrations instead. One heterozygous variant of uncertain significance was detected in two of the samples that were screened negative. There were no second variants detected in these cases. The third sample contained no reportable variants. Overall, these results confirmed that the current cutoffs used in the study were able to minimize the false negative, but additional large-scale studies with long-term follow ups would be very helpful to determine the sensitivity and specificity of the assay for all four conditions.

Our assay is currently limited in that low or reduced peptide concentrations may be caused by experimental errors, such as specimen spots not uniformly saturated with blood, poorly collected and improperly dried specimens, or the health conditions of each newborn, as premature infants or sick/transfused infants may have different protein levels from healthy and at-term infants. We still need to confirm each case with reduced or absent protein concentrations by standard diagnostic methods such as DNA sequencing. Similarly, negative assay results cannot definitively rule out the disease. The primary initial challenge in expanding this method into routine newborn screening could be the high cost of instruments with advanced sensitivity. However, we believe this cost would be offset by the long-term benefits of the early detection and preemptive treatment, which could significantly reduce overall healthcare expenses.

In summary, we have developed a highly sensitive and rapid proteomics-based assay for the direct quantification of extremely-low-abundance proteins in DBS samples. Our assay directly targets the proteins that are affected by genetic mutations causing WAS, XLA, and ADA deficiency, and simultaneously quantifies target proteins to screen for all four disorders. This provides a fast and effective approach for screening for these conditions, facilitating earlier clinical diagnosis and follow-up treatment, and subsequently improves patients’ prognoses.

## 5. Conclusions

Newborn screening has rapidly become one of the most successful public health programs in the US by changing the clinical course of thousands of patients’ lives. A novel LC-MS/MS proteomics-based screening test has been developed for the early detection of four rare genetic diseases in newborns. These are life-threatening conditions with preventable severe complications when diagnosed early in life. Our novel proteomic assay has the sensitivity and precision required to quantify low-abundance proteins from dried blood spots that enable us to detect these four life-altering diseases, and can accurately differentiate between patients and unaffected individuals with a sample runtime of less than 3 min. The assay also yields reproducible results in different test sites, making it the appropriate screening candidate for countries across the globe, and has the potential to be expanded to include other rare diseases.

## Figures and Tables

**Figure 1 IJNS-11-00006-f001:**
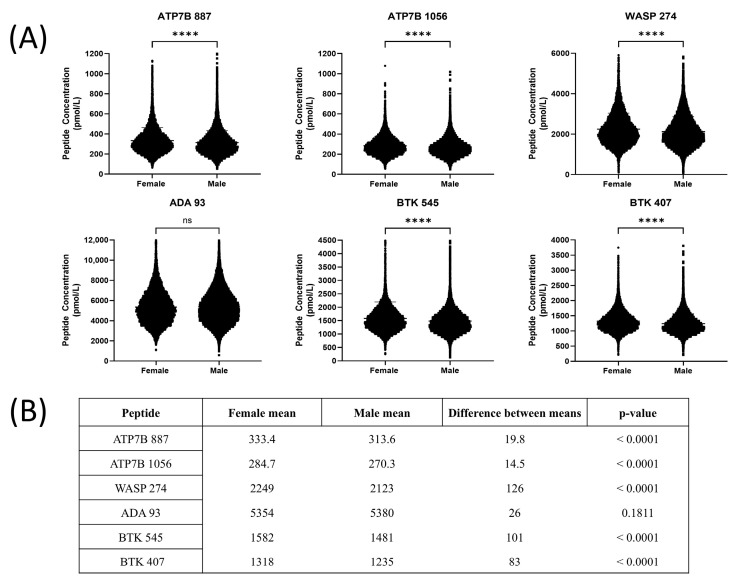
(**A**) Peptide concentrations between genders (male/female) (**** = *p* ≤ 0.0001, ns = no significance). (**B**) Mean peptide concentrations between genders, differences between the means and significance.

**Figure 2 IJNS-11-00006-f002:**
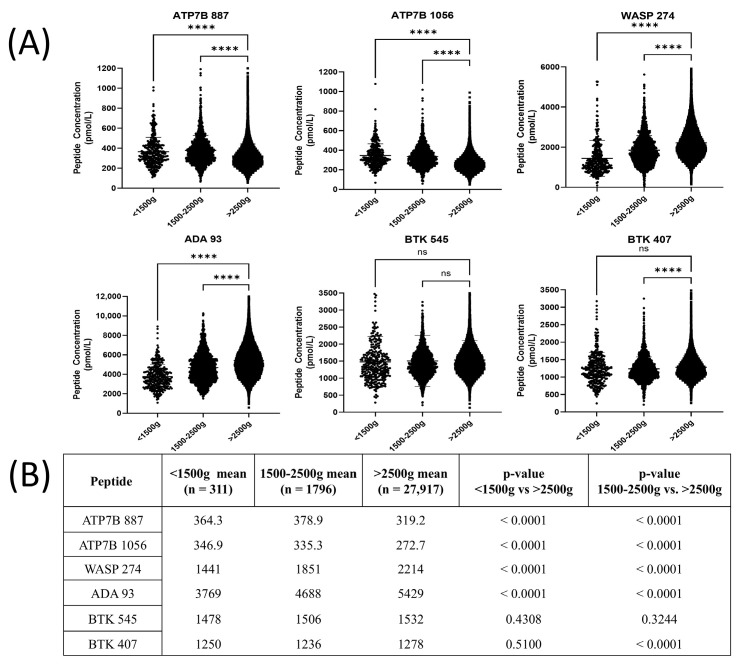
(**A**) Peptide concentrations between different birth weight groups (<1500 g, 1500 g–2500 g, and >2500 g) (**** = *p* ≤ 0.0001, ns = no significance). (**B**) Statistical significances in the average peptide concentrations between different birth weight groups (<1500 g, 1500 g–2500 g, and >2500 g). The *p*-value was calculated using the Welch ANOVA test between the <1500 g and >2500 g groups, and 1500 g–2500 g and >2500 g groups.

**Figure 3 IJNS-11-00006-f003:**
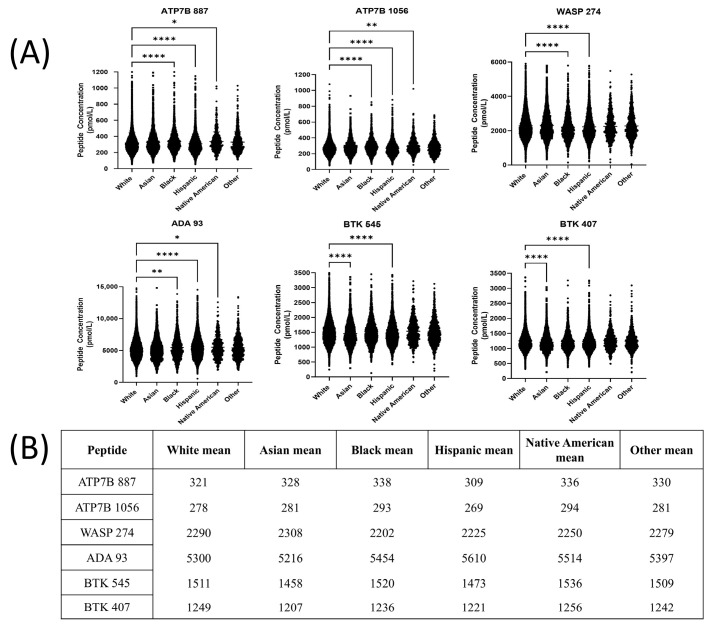
(**A**) Peptide concentrations between ethnicities (* = *p* ≤ 0.05, ** = *p* ≤ 0.01, **** = *p* ≤ 0.0001). (**B**) Mean concentration differences between ethnicities in pmol/L.

**Figure 4 IJNS-11-00006-f004:**
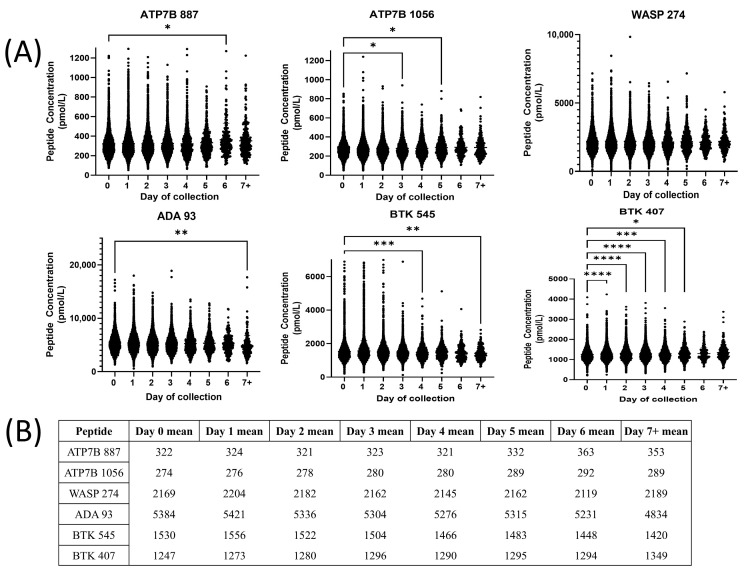
(**A**) Peptide concentrations in different groups of age of newborn sample collection in days (* = *p* ≤ 0.05, ** = *p* ≤ 0.01, *** = *p* ≤ 0.001, **** = *p* ≤ 0.0001). (**B**) Mean peptide concentrations in different age groups in pmol/L.

**Table 1 IJNS-11-00006-t001:** The limit of blank, limit of detection, and limit of quantification in pmol/L for all six target peptides.

Detection Capability (pmol/L)	ATP7B 887	ATP7B 1056	WASP 274	ADA 93	BTK 407	BTK 545
Limit of Blank	15.5	13.7	6.2	77.5	7.5	2.6
Limit of Detection	24.5	22.7	23.9	172.7	21.4	13.3
Limit of Quantification	40.2	49.6	51.4	197.8	41.2	108.8

**Table 2 IJNS-11-00006-t002:** Carryover measured in the blank samples ran immediately after the samples containing high concentration of target peptides.

Target Peptide	Carryover (%)
ATP7B 887	0.2–4.1
ATP7B 1056	0.9–4.2
WASP 274	0.1–0.6
ADA 93	5.0–21.2
BTK 545	0.4–2.0
BTK 407	0.0–0.8

**Table 3 IJNS-11-00006-t003:** Proposed diagnostic cutoff for each peptide based on initial range testing.

Peptide Target	Mean (pmol/L)	Median (pmol/L)	SD	Initial Cutoff (pmol/L)
ATP7B 887	273.1	266.2	80.1	66.6
ATP7B 1056	275.0	269.8	75.1	67.5
WASP 274	2291.9	2263.7	575.9	226.4
ADA 93	5800.0	5728.2	1543.0	1145.6
BTK 407	1148.7	1122.1	225.7	112.2
BTK 545	1599.1	1563.7	342.3	156.4

**Table 4 IJNS-11-00006-t004:** Differences in peptide concentration between day 0 (initial) and day 5 (final) extracted from DBS cards stored at 10 °C, 22 °C, and 37 °C.

Peptide	10 °C	22 °C	37 °C
	% Difference	% Difference	% Difference
ATP7B 887	−6.4	−6.2	−6.5
ATP7B 1056	−10.5	0.1	−4.9
WASP 274	−22.5	−22.8	−21.4
ADA 93	6.3	−7.1	2.0
BTK 545	4.1	−2.2	3.1
BTK 407	−2.1	11.3	2.6

**Table 5 IJNS-11-00006-t005:** Demographic information for the 30,024 de-identified newborn samples received from the Washington State Department of Health Newborn Screening Laboratory.

**(A) Gender and Birth Weight**		
**Category**	**Number**	**%**
Male	14,548	48.5
Female	15,476	51.5
<1500 g BW	311	1.0
1500–2500 g BW	1796	6.0
>2500 g	27,917	93.0
**(B) Ethnic Background**		
**Ethnicity**	**Number**	**%**
White	16,104	58.6
Hispanic	5109	18.6
Asian	2879	10.5
Black	1906	6.9
Native American	560	2.0
Other	916	3.3
**(C) Date of Collection After Birth**		
**Date of Collection After Birth**	**Number**	**%**
0 day	4845	16.1
1 day	12,361	41.2
2 days	6617	22.0
3 days	3836	12.8
4 days	1352	4.5
5 days	567	1.9
6–14 days	446	1.5

**Table 6 IJNS-11-00006-t006:** Diagnostic cutoffs determined for each target peptide on both the ESI low-flow probe and the ionKey configuration.

ESI Low Flow						
	**ATP7B 887**	**ATP7B 1056**	**WASP 274**	**ADA 93**	**BTK 545**	**BTK 407**
MEDIAN	279.5	255.1	2104.5	5805.7	1604.7	1163.6
SD	96.0	81.2	844.9	1624.1	447.8	289.9
CV	34.4	31.8	40.1	28.0	27.9	24.9
Number	19,524	19,526	19,526	19,526	18,114	19,526
CUTOFF	78.3	71.4	210.5	1161.1	160.5	116.4
**IonKey**						
	**ATP7B 887**	**ATP7B 1056**	**WASP 274**	**ADA 93**	**BTK 545**	**BTK 407**
MEDIAN	359.8	292.4	2018.8	4189.4	1211.2	1299.9
SD	149.7	99.2	599	1297.7	721.9	322.3
CV	41.6	33.9	29.7	30.5	59.6	24.8
Number	10,488	10,498	10,498	10,498	10,498	10,498
CUTOFF	64.8	52.6	201.9	837.9	121.2	130.0

**Table 7 IJNS-11-00006-t007:** True positive sample screened for Wilson Disease and presumptive positive samples for Wilson Disease, Wiskott-Aldrich Syndrome, ADA deficiency, and X-linked agammaglobulinemia. The specimen age (days), body weight (g), sex, clinical diagnosis, and MS system used to analyze each sample and genotype are shown. Concentrations of peptides used for the screening of each disease are bolded.

**True Positive**												
Sample	Specimen Age (Days)	BW (g)	Sex	Diagnosis	MS System	ATP7B 887	ATP 1056	WASP 274	ADA 93	BTK 545	BTK 407	GENOTYPE
1	1	4105	M	WD	ESI Low Flow	**71.0**	**96.5**	1782.7	5169.5	1467.8	1677.9	p.Pro610Leu/ p.Arg1224Leu
**False Positive**												
Sample	Specimen Age (Days)	BW (g)	Sex	Diagnosis	MS System	ATP7B887	ATP1056	WASP274	ADA93	BTK 545	BTK 407	GENOTYPE
1	2	3690	M	WD	ionKey	**67.6**	**134.5**	2569.4	3452.7	823.3	895.5	c.3402del
2	2	3870	M	WD	ESI Low Flow	**66.4**	**70.0**	2193.3	7598.3	2030.5	1293.1	NO VARIANTS
3	2	3840	F	WD	ESI Low Flow	**64.3**	**59.2**	1307.1	2414.2	594.8	470.3	p.Met33Thr
4	1	2290	M	WAS	ionKey	248.7	224.1	**144.0**	2750.5	506.9	402.3	NO VARIANTS
5	4	1760	M	WAS	ionKey	345.1	506.2	**155.1**	2752.8	728.0	656.2	INSUFF DNA
6	0	1550	M	WAS	ESI Low Flow	126.6	188.1	**39.9**	2512.8	206.3	216.9	INSUFF DNA
7	0	1800	M	WAS	ESI Low Flow	182.0	235.7	**47.2**	3068.1	279.8	206.9	INSUFF DNA
8	1	2067	M	WAS	ESI Low Flow	265.2	284.2	**192.4**	4348.9	605.9	410.0	INSUFF DNA
9	1	1090	F	ADAD	ionKey	583.6	1078.5	1291.3	**1078.1**	1363.6	835.5	p.Gly94Asp
10	2	3175	F	ADAD	ionKey	498.4	427.5	1690.0	**1129.5**	971.9	1101.7	NO VARIANTS
11	1	2740	M	ADAD	ionKey	567.9	511.4	2572.0	**1186.2**	1207.9	1308.9	p.Ala215Thr
12	3	3530	M	XLA	ESI Low Flow	190.4	90.5	762.0	4701.1	**129.0**	**369.1**	NO VARIANTS

**Table 8 IJNS-11-00006-t008:** Potential false negative study results from the pilot study sample cohort. The gender, peptide concentration (pmol/L), sequencing report results, and diagnostic conclusion are shown.

Case	Gene	Gender	ATP7B 887	ATP7B 1056	Sequencing	Conclusion
1	ATP7B	F	141.5	59.3	p.Trp779Gly	Likely pathogenic; no second variant
2		M	88.3	70.4	p.Ile161Thr/p.Leu1015=	One VUS; second benign
3		M	107.6	86.7	p.Leu1015=	Benign
4		M	73.6	57.0	p.Gly256Ala/p.Leu1015=	One pathogenic and one benign
5		M	97.5	90.6	No variants detected	Negative
6		M	156.0	56.3	No variants detected	Negative
7		F	99.5	85.8	p.Arg919Trp	Likely pathogenic. No second variant
8		F	134.3	65.6	p.Asn687Ile	One VUS; no second variant
9		M	105.1	85.0	No variants detected	Negative
10		M	111.0	85.0	c.-370C>A	One VUS; no second variant
11		F	89.2	167.6	No variants detected	Negative
12		F	88.0	239.3	No variants detected	Negative
13		M	80.3	296.5	p.Gly367Asp	One VUS; no second variant
14		M	69.5	97.6	p.Ala971Val	One VUS; no second variant
15		F	78.1	95	c.2009_2015de	Pathogenic variant; no second variant
16		M	80.2	183.1	c.3402del	Pathogenic variant; no second variant
17		F	83.4	151	p.Pro461Ser	One VUS; no second variant
18		M	80.8	115.4	p.Thr977Met	Pathogenic variant; no second variant
19		F	81.7	110.6	p.Val1262Phe	Likely pathogenic; no second variant
20		M	79.7	100.8	c.2304dup	Pathogenic variant; no second variant
21		F	77	147.1	p.Ile747=	One VUS; no second variant
22		M	86.9	136.9	p.Pro539Leu/p.Ser876Cys	One likely pathogenic and one VUS
23		F	81.4	122.8	No variants detected	Negative
**Case**	**Gene**	**Gender**	**WASP 274**		**Sequencing**	**Conclusion**
24	WASP	M	227.3		No variants detected	Negative
25		M	281.6		No variants detected	Negative
26		M	328.5		No variants detected	Negative
27		M	352.1		No variants detected	Negative
28		M	381.4		No variants detected	Negative
**Case**	**Gene**	**Gender**	**ADA 93**		**Sequencing**	**Conclusion**
30	ADA	F	1129.5		No variants detected	Negative
31		M	1186.2		p.Ala215Thr	One VUS; no second variant

## Data Availability

Data are available upon request.

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
