# Peer review of "Advancing Newborn Screening in Washington State: A Novel Multiplexed LC-MS/MS Proteomic Assay for Wilson Disease and Inborn Errors of Immunity"

_2409-515X, 2025, doi:10.3390/ijns11010006_

Round 1

Reviewer 1 Report

Comments and Suggestions for Authors

Introduction - can you please reorganize the disease information section?  For example, page 2, line 73/beginning of IEI descriptions would be better placed as a new paragraph.  

Page 3, line 102 - I think it would be helpful to the reader to state that you are analyzing 6 peptides here.

Page 3, line 112 - I was confused by number of peptides here.  Maybe reword to clarify that WD has 2 peptides, XLA has two peptides.... the first time I read it I thought that the two peptides referred to two single peptides that are specific for both WD and XLA.

Page 3, line 118.  What is the significance of using fish blood for validation?  Can this be explained for non-laboratorians?

2.1.2 Interference - I think readers would appreciate a list (or even general description) of interfering compounds in the text vs having to go to the supp info.   

Author Response

please see the attachement.

Reviewer 2 Report

Comments and Suggestions for Authors

The authors present an interesting and well-written paper, presenting data supporting a novel approach to screening newborns for four conditions: WD, XLA, ADA deficiency and WAS using LC-MS/MS. They provide data from their pilot study, and a prospective study on over 30,000 anonymised newborn DBS.

My comments are below:

-              Abstract

o   line 26-27 ‘four of them had insufficient DNA’ add ‘for testing’ or ‘for confirmation’

-              Introduction

o   Line 47 ‘highly diagnostic’ – modify phrasing

o   Discussion of diseases included (line 66 onwards) – this section requires review and consider reformatting to discuss each condition in a new paragraph for clarity

o   Specific recommended changes – line 76 ‘wholly absent immune system’ – remove this line, as this is an incorrect description. One cannot have an ‘absent’ immune system, of which there are many components and compartments (see IUIS classification of IEI for further background: https://iuis.org/committees/iei/ see table; and https://pubmed.ncbi.nlm.nih.gov/35748970/) – these diseases are classified according to their predominant immunological defect

o   Line 75 – change to ‘over 485 genetic disorders’ and reference above paper

o   Line 78 – a better description of XLA is that it results in absent or very low B cells, panhypogammaglobulinaemia and recurrent infections

o   It would be worth acknowledging briefly that there is another method for screening for XLA and other congenital B cell deficiencies, using KREC measurement. I do not believe that this is used within North America, however it is used in several other countries, typically as a multiplexed assay along with TREC (and SMA)

o   Line 81: It should be stated that WAS is a combined immunodeficiency disease, hence resulting in recurrent and severe infections, along with bleeding complications. Thrombocytopaenia and reduced platelet volume are key features

o   Line 80, please note ‘replacement immunoglobulin’ and ‘gamma globulin’ are the same thing. Recommend using the term ‘immunoglobulin replacement therapy’ which is standard terminology

o   Line 84, ADA deficiency. It is important to recognise and mention that this causes a severe combined immunodeficiency (SCID) phenotype. As highlighted, there are some late onset forms but classically it presents in infancy and affected individuals typically have absence of T, B and NK cells

o   Line 85: ‘immunodeficiency characterised by the disrupted function of ADA’ – please revise or remove this sentence, providing an improved descriptor as to this disease as above

o   Line 87: state that the currently available screening method for ADA deficiency and other forms of SCID is using TREC measurement and briefly describe this

o   Line 89: gene therapy (GT) is another curative therapy for ADA deficiency SCID and should be mentioned. Note that GT and HSCT are curative options (enzyme replacement is generally for short-term use)

o   Line 92: ‘chronic negative sequelae’ – suggest change to ‘and other severe disease complications’

o   Line 93: it should be pointed out that the TREC assay will only identify those IEI which manifest with low or absent T cells (i.e. it does not screen for all forms of IEI)

o   Line 94-95: suggest removing this line, will be covered as above, and second tier and diagnostic testing is generally reflexed after a positive screen, so I would disagree that diagnosis will be delayed and there could be complications whilst awaiting this as empiric treatment is started in the interim

o   Line 96-100: recommend removing, as what is mentioned here is what occurs in the absence of screening for SCID, for reasons explained above re empiric treatment whilst completing work-up. I suspect that here you are trying to justify why an ADA-deficiency specific assay could be helpful, but you would need to express that alongside routine TREC screening, your assay will help identify those infants who have ADA-SCID specifically (as opposed to other molecular causes of SCID, of which there are several)

o   Line 101: please include a description in the interaction of the biomarkers you are assessing for each condition, some background on each, and why and how they were selected. There is no information provided regarding this before going into methods, and this will help structure the paper and better orient the reader

-              Materials & Methods

o   Line 117: What was the purpose of using fish blood mixed with human blood for validation studies? Was whole blood used, or was it blotted onto filter paper for this stage of the validation?

o   Line 149: change ‘ran’ to ‘run’

o   Line 164: what was the breakdown of the diseases in the 49 positive controls?

o   Line 169: suggest change ‘blinded positives’ terminology and refer to these as positive controls and discuss how these were all run blinded

o   Line 173 ‘presumed negative’ – do you mean presumed healthy newborns?

o   Section 2.2 pilot study – typographical - check correct tense used throughout

o   Section 3.1.5 – please clarify whether levels below or above cut-off are abnormal – this is not immediately clear

o   Section 3.1.5 – if these assays detect carrier status, how will this be managed? If this were to be implemented routinely, will carrier status be reported? Address in discussion.

o   Section 3.2.4 – ‘presumptive positives’ – I presume this refers to those who had an abnormal screening test, genetic variant identified but unable to definitively make the diagnosis as the samples were all de-identified, so you have been unable to correlate with clinical/other laboratory findings? And this is the same for ‘false negatives’, and in table 8 ‘potential false negative’? I would suggest explaining more clearly what you mean by each of these descriptors, and using the decided terminology consistently throughout for clarity

-              Discussion

o   Line 460: ‘positive blind samples’ – needs rephrasing as discussed earlier

o   It is noted that there were statistically significant differences in peptide concentrations based on BW, gender, ethnicity but the authors assess that these differences are small and not significant enough to necessitate specific cut-offs for different patient groups. It is discussed that there is variability in BTK peptides over time – is a different cut-off applied for those samples processed during the second screening (7-14 days) for these, and any of the other peptides?

o   Line 525: reported in ‘the’ literature

o   Line 532: remove ‘from’

o   Line 538: I would disagree that the TREC assay has a high ‘false positive’ rate. This is a test for T cell lymphopaenia, for which SCID is the target condition, but due to the nature of the assay, newborns with other conditions will be identified. I would suggest moving this line, or explaining it in further detail to clarify.

o   What are the future directions – is there a plan to evaluate this with identifiable patients to support the findings with full clinical/diagnostic data? What are the next steps?

o   Summary – line 570 states that this is a cost-effective approach, however there is no prior presentation of any data where a health economic analysis has been undertaken – how has this conclusion been reached?

o   Line 570-572 requires rewording – consider ‘this provides a fast and effective approach to screening for these conditions, facilitating earlier diagnosis and treatment’

-              Appendix

o   Line 638: reword ‘genetically confirmed positive blind samples’, as discussed earlier and use consistent terminology throughout

-              Supplementary data

o   Figure S2: time of ‘receipt’ rather than ‘receival’

o   Table S5, site 3 case 16: variant 1 ‘based on family history’ but variant not given? Is this data not available? Please clarify

o   Table S6: ‘False negative study…’ – please modify terminology for clarity throughout

Reviewer 3 Report

Comments and Suggestions for Authors

Thank you for allowing me to review this study that describes the use of a proteomic Immuno-SRM multiplex assasy to target 4 proteins to screen for Wilsons disease and 3 inborn errors of immunity.

I have the following comment:

The validation of the assay of the method is appropriate and no changes are recommended.

For the pilot study explanation (line 198-219), I did not understand the abbreviations used in lines 205-206. e.g 45 CFR 46.404/21 CFR 50.51) and IDE exempt. 

The results of the clinical validation of the study are fully described, although it was hard to follow how this affected the cut-offs for the pilot study. For example the experience of using ATP7B 1056 AND 887 at 2 different sites were different although they were different cases. It would be good to discuss what these discordant findings meant for the pilot study (perhaps in the discussion). This clinical validation section may benefit from tabulated results.

The pilot study identified 1 confirmed case of WD and 12 FP results for WD and IEIs in 30024 bloodspots. I would suggest a paragraph on challenges of expanding the method into routine newborn screening as I assume this will be the ultimate goal. 
